# OMICS and Other Advanced Technologies in Mycological Applications

**DOI:** 10.3390/jof9060688

**Published:** 2023-06-19

**Authors:** Nalin N. Wijayawardene, Nattawut Boonyuen, Chathuranga B. Ranaweera, Heethaka K. S. de Zoysa, Rasanie E. Padmathilake, Faarah Nifla, Dong-Qin Dai, Yanxia Liu, Nakarin Suwannarach, Jaturong Kumla, Thushara C. Bamunuarachchige, Huan-Huan Chen

**Affiliations:** 1Centre for Yunnan Plateau Biological Resources Protection and Utilization, College of Biological Resource and Food Engineering, Qujing Normal University, Qujing 655011, China; nalinwijayawardene@yahoo.com; 2Department of Bioprocess Technology, Faculty of Technology, Rajarata University of Sri Lanka, Mihintale 50300, Sri Lanka; ksdezoys@tec.rjt.ac.lk (H.K.S.d.Z.); faaranifla@gmail.com (F.N.); tcbamunu@tec.rjt.ac.lk (T.C.B.); 3Section of Genetics, Institute for Research and Development in Health and Social Care, No: 393/3, Lily Avenue, Off Robert Gunawardane Mawatha, Battaramulla 10120, Sri Lanka; 4National Center for Genetic Engineering and Biotechnology (BIOTEC), National Science and Technology Development Agency (NSTDA), 111 Thailand Science Park, Phahonyothin Road, Khlong Nueng, Khlong Luang, Pathum Thani 12120, Thailand; nattawut@biotec.or.th; 5Department of Medical Laboratory Sciences, Faculty of Allied Health Sciences, General Sir John Kotelawala Defence University Sri Lanka, Kandawala Road, Rathmalana 10390, Sri Lanka; cbr2704@kdu.ac.lk; 6Department of Plant Sciences, Faculty of Agriculture, Rajarata University of Sri Lanka, Pulliyankulama, Anuradhapura 50000, Sri Lanka; rasaniep@gmail.com; 7Guizhou Academy of Tobacco Science, No.29, Longtanba Road, Guanshanhu District, Guiyang 550000, China; liuyanxia306@163.com; 8Research Center of Microbial Diversity and Sustainable Utilization, Faculty of Science, Chiang Mai University, Chiang Mai 50200, Thailand; suwan.462@gmail.com (N.S.); jaturong_yai@hotmail.com (J.K.); 9Department of Biology, Faculty of Science, Chiang Mai University, Chiang Mai 50200, Thailand; 10Key Laboratory of Insect-Pollinator Biology of Ministry of Agriculture and Rural Affairs, Institute of Agricultural Research, Chinese Academy of Agricultural Sciences, Beijing 100193, China

**Keywords:** fungal taxonomy, high-throughput sequencing, multiomics approaches, dark taxa, phylogenetic analysis, fungal metabolites

## Abstract

Fungi play many roles in different ecosystems. The precise identification of fungi is important in different aspects. Historically, they were identified based on morphological characteristics, but technological advancements such as polymerase chain reaction (PCR) and DNA sequencing now enable more accurate identification and taxonomy, and higher-level classifications. However, some species, referred to as “dark taxa”, lack distinct physical features that makes their identification challenging. High-throughput sequencing and metagenomics of environmental samples provide a solution to identifying new lineages of fungi. This paper discusses different approaches to taxonomy, including PCR amplification and sequencing of rDNA, multi-loci phylogenetic analyses, and the importance of various omics (large-scale molecular) techniques for understanding fungal applications. The use of proteomics, transcriptomics, metatranscriptomics, metabolomics, and interactomics provides a comprehensive understanding of fungi. These advanced technologies are critical for expanding the knowledge of the Kingdom of Fungi, including its impact on food safety and security, edible mushrooms foodomics, fungal secondary metabolites, mycotoxin-producing fungi, and biomedical and therapeutic applications, including antifungal drugs and drug resistance, and fungal omics data for novel drug development. The paper also highlights the importance of exploring fungi from extreme environments and understudied areas to identify novel lineages in the fungal dark taxa.

## 1. Introduction

Fungi comprise diverse taxa that are abundant in various environments. They have crucial ecological roles as decomposers, mutualists, and disease-causing agents [1,2,3]. Taxonomic studies, i.e., describing fungi, have been carried out since the 18th century, focusing on macro- and micro-morphological characteristics (or phenotypic) and host–fungal relationships [4,5]. Due to the morphological plasticity, mycologists faced a great challenge in identifying new species prior to the availability of molecular methods such as Polymerase Chain Reaction (PCR) and DNA sequencing. Despite the rudimentary understanding of fungi in the past, traditional fungal taxonomy was important for the characterization of fungi used in various industries, particularly in the food and beverage industry [6].

Recent advances in technology have revolutionized our understanding of the Kingdom of Fungi. Fungal taxonomists are now able to use molecular methods in addition to morphology for species identification and classification, which have enabled a more comprehensive understanding of the fundamental biology and taxonomy of fungi [7]. However, it has been recognized that many of the industrially important species are cryptic species that occur as species complexes, such as *Aspergillus* and *Penicillium*. Moreover, many species lack visible distinguishing physical characteristics such as fruiting bodies (ascomata, conidiomata) or conidiophores [5,8]. According to Wang and team [9], the majority of known fungal classifications do not possess distinctive physical structures that can be described. These classifications are referred to as “dark taxa” [10,11]. Identifying these taxa is crucial in accurately estimating the current species diversity of fungi, which is believed to range between 2.2 million and 3.8 million species based on the host–fungi index [12].

The use of high-throughput sequencing (HTS) and metagenomics in analyzing DNA from environmental samples has become crucial in the identification of new fungal lineages [13]. In addition, the development of other omics technologies interrogating different cellular components and the use of these technologies in various polyphasic methods have significantly advanced research in areas such as biodiversity, physiological ecology, environmental sciences, and natural product biosynthesis [14]. Among these omics approaches, proteomics [15,16], transcriptomics [17], metatranscriptomics [18], and metabolomics [19] have revolutionized the current understanding of the biological processes of fungi. In addition, more specialized omics methods such as ionomics, glycomics, glycoproteomics, glycogenomics [20], lipidomics [21,22], and interactomics [23,24] coupled with bioinformatics [25,26] can contribute to a greater understanding of fungal metabolism. The combination of omics approaches (multiomics) can be used to characterize fungal genomes and their metabolites, making multiomics approaches essential for detecting and characterizing novel metabolites with important biological properties, such as anticancer, antimicrobial, and antidiabetic for human health applications [27,28,29]. In this review article, we compile recent progress in the taxonomy of fungi and the development of the food and pharmaceutical industries since the adoption of omics technologies.

## 2. Approaches in Taxonomy and Classification

### 2.1. Genetic Data and Phylogenetic Analysis

In the last three decades, the use of PCR and Sanger sequencing of rDNA has led to a rapid increase in the understanding of mycological taxonomy [7]. The rDNA unit comprising the 18S (small subunit—SSU) and 28S (large subunit—LSU) ribosomal RNA genes, along with the internal transcribed spacers (ITS1 and ITS2), provides the sequence information necessary for genus-level resolution in phylogenetic analyses. The ITS1–2 non-coding region is particularly useful for resolving relationships among closely related fungal taxa and for species identification, as it has a high degree of variability among fungi. This region is recommended as the primary fungal barcode, or as a single locus for phylogenetic analysis [30]. In addition to ITSs, protein-coding genes including β-tubulin (*BenA* or *Tub2*), translational elongation factor 1 alpha (*TEF-1* α), RNA polymerase II second largest subunit (*RPB1* and *RPB2*), calmodulin (*CAM* or *CAL*), and mtLSU are used as secondary barcoding markers to improve the resolution of phylogenetic relationships in fungi. It is also important to note that the selection of barcode markers should be taxon-specific and depend on the phylogenetic group and the question to be answered [31,32,33,34]. The use of multiple markers in the polyphasic taxonomic approach provides a more complete picture of evolutionary relationships among fungi and allows for phylogenetic reconstruction at different levels of fungal taxonomy. This approach has been particularly successful for resolving the Ascomycota, Zygomycota, and Basidiomycota [30,35,36]. An example of using a polyphasic taxonomic method based on fungal molecular markers, morphology, and phylogenetic data (ITS, LSU, SSU, *RPB2*, *TUB*, and *TEF-1 α*) as well as whole-genome sequencing (WGS), recently led to the discovery of three new fungal taxa isolated on surfaces associated with NASA spacecraft assembly facilities [37].

Performing multi-loci phylogenetic trees is highly recommended in modern mycology [30,35,38]. Recent studies based on multi-gene regions have led to higher species resolutions of speciose genera, such as *Aspergillus* [39,40], as well as *Penicillium* [41]. Furthermore, phylogenomic studies, which are based on the entire genome, are viewed as a powerful tool for future mycology research [42,43,44], the study of groups that are highly plastic and unculturable, but vital in industries such as agriculture [45]. Phylogenomics using fungarium specimens led to the revision of a family within the Agaricales [46], underlining the power of this approach [47,48]. In the research [49], a hybrid approach of low-coverage genome sequencing and multi-gene phylogenetics was utilized to elucidate the intricate Mortierellaceae phylogeny. This methodology resulted in the discovery and proposal of seven novel genera, thereby establishing a more distinct taxonomic structure for the family. The outcomes of this study enhance our comprehension of Mortierellaceae’s diversity, biology, and evolution, facilitating more focused and comprehensive investigations going forward.

Phylogenomic studies utilizing genome-scale data have significantly enhanced our understanding of the tree of life. This research focuses on the poorly resolved evolutionary relationships within major fungal lineages near the base of the fungal phylogeny. By compiling a comprehensive dataset of 1644 species and 290 genes, various analyses were conducted to construct a robust phylogeny of the fungal kingdom, uncovering historically unresolved relationships and highlighting episodes of ancient diversification. The findings provide a solid foundation for exploring fungal evolution and offer valuable insights for future phylogenetic and taxonomic investigations in the field [50]. In [51], this study utilized genome-scale molecular phylogenetics to accurately classify and reconstruct the evolutionary relationships within *Aspergillus*. By analyzing a comprehensive dataset of 711 fungal genomes and employing a set of 1362 molecular markers, taxonomic controversies were resolved, misidentifications were identified, and new lineages were discovered. These findings highlight the power of phylogenomics in achieving precise taxonomic classifications and shedding light on the evolutionary history of important genera [50,51].

Unculturable taxa represent a significant portion of the Kingdom Fungi, potentially representing missing lineages that can be analyzed from ecological samples using HTS. In addition, HTS technologies are proving valuable in analyzing DNA from older herbarium specimens that lack DNA sequences from the PCR/Sanger sequencing approach [52].

### 2.2. Discovering New Taxa in Known Lineages

Fungi are used as cell factories to produce antibiotics, enzymes, and other compounds for diverse industrial uses [53,54]. The development of fungal cell factories was driven by mycological research employing integrated omics technologies, which has deepened our understanding of fungal taxa at the biological, biochemical, and biophysical level [55,56,57,58]. Multiomics refers to the holistic approach of analyzing multiple types of data, such as the genetic information encoded in the metagenome and the proteins present in the metaproteome. Multiomics can also be extended to interconnected systems, such as the microbial metagenome in conjunction with the host’s metabolome [59].

For instance, these omics techniques can be used in mycological synthetic biology and fungal biotechnology applications such as fungal gene function [60], fungal survival under temperature stress [61], *Candida albicans* biofilm formation and drug resistance [62], and fungal communities in forest soils [63]. The integration of cutting-edge sequencing technologies and bioinformatics techniques in mycological research has led to a marked advancement in the field. The ongoing 1000 Fungal Genomes Project (https://1000.fungalgenomes.org, accessed on 11 January 2023) serves as a prime example of this progress, with a significant portion of the targeted fungal genomes sequenced and annotated [64]. This project has led to a deeper understanding of fungal biology, evolution, pathogenicity, and ecology [64], while providing the understanding of fungal interactions with their environments [65]. For example, these insights provide a foundation for population structure, genetic diversity, and putative ecological drivers of clinically relevant fungi [16,66].

In recent years, the use of omics techniques has enabled the identification of novel fungal species based on their genetic and protein profiles. New fungal species have been described using omics techniques such as phylogenomic–morphological analyses [67], and proteomics techniques such as matrix-assisted laser desorption/ionization time-of-flight mass spectrometry (MALDI-TOF MS) combined with phenotypic and DNA sequence data [68].

Besides DNA, proteins are a valuable information source for fungal systematics. Protein profiles can be obtained in a convenient, quick, accurate, and cost-effective manner using MALDI-TOF MS and protein fingerprinting [58]. MALDI-TOF MS has been used for the identification of aquatic hyphomycetous taxa [69], *Aspergillus*, *Fusarium*, Mucorales species [70], fungal dermatophytes [71], polyporoid/hymenochaetoid mushrooms [72], black-yeast-like fungi, and wine yeast [73,74]. Furthermore, unique proteins can be used to classify fungi as new species/genera and have great potential for solving many unknowns in fungal systematics [58]. For example, *Hypomontagnella* is a new genus in *Hypoxylaceae* that is accommodated along with other taxa that produce sporothriolide antifungal polyketides.

### 2.3. Proteomics in Fungal Systematics

Proteomics is an essential component of fungal omics technologies, providing valuable insights into the protein expression, modifications, and interactions within fungal systems. By utilizing proteomics approaches, such as mass-spectrometry-based techniques, researchers can identify and quantify the proteins present in fungal cells or tissues [72,73,74,75,76,77]. Proteomics enables the characterization of fungal proteomes, including the identification of novel proteins, post-translational modifications, and protein–protein interactions. This information helps elucidate the functional roles of proteins in fungal biology, including enzymatic activities, cellular processes, and signaling pathways. Integrating proteomics with other omics data, such as genomics and transcriptomics, allows for a more comprehensive understanding of fungal systems and their responses to environmental stimuli or pathogenic interactions. Additionally, proteomic analyses contribute to the discovery of potential targets for antifungal drugs and the development of proteomic-based diagnostics or therapeutics [20,72,73,74,75,76,77]. As proteomics technologies continue to advance, including improvements in sensitivity, throughput, and data analysis, they will play an increasingly significant role in unraveling the complexities of fungal biology and advancing applications in various fields. Proteomics plays a significant role in fungal systematics by providing valuable insights into species identification, phylogenetic analysis, biomarker discovery, fungal barcoding, comparative proteomics, and functional analysis. Proteomic analysis enables the identification and classification of fungal species by establishing protein-based fingerprints. It contributes to evolutionary studies by comparing protein expression patterns and constructing phylogenetic trees. Proteomics also aids in the discovery of species-specific biomarkers for rapid identification, complements DNA barcoding approaches, and helps resolve taxonomic ambiguities [16,78,79]. Comparative proteomics reveals molecular mechanisms underlying phenotypic variation and adaptation in fungi. Additionally, proteomics enables functional characterization of fungal proteins, enhancing our understanding of fungal biology. Integrating proteomic data with other molecular information advances fungal systematics, leading to a comprehensive understanding of fungal diversity and biology [79,80].

### 2.4. Novel Lineages of Dark Taxa from Understudied Geographical Regions

Approximately 160,000 species of fungi have been named to date (Species Fungorum 2023; accessed on 11 January 2023), which is far fewer than the estimated total. There are many areas of the world that remain understudied and are likely to harbor a wealth of new fungal lineages and species. For example, fungal diversity is under-explored in tropical rainforests, deep sea, marine and semi-marine, and extreme ecosystems, i.e., karst caves and polar environments [81,82,83,84,85,86,87]. Moreover, based on pyrosequencing of soil-inhabiting fungi, Tedersoo et al. [88] suggested an abundance of undescribed taxa in Rozellomycota (Cryptomycota), Ascomycota, and Basidiomycota. It is also thought that host plants in understudied geographic regions are a rich source of novel parasitic fungal taxa [75]. 

Understanding the intricate biology, ecology, and evolution of fungi heavily relies on the discovery and classification of these new fungal lineages [89,90,91]. Such efforts play a crucial role in enhancing our comprehension of the evolutionary relationships and overall diversity among fungi. For instance, investigations into the roles of rock-inhabiting fungi and ectomycorrhizal fungi contribute to our understanding of global biogeochemical cycles [92,93,94,95]. Moreover, through the application of phylogenomic, phylogenetic, and morphological analyses, researchers can uncover not only new fungal species but also their associated lineages from different countries [91,96,97]. These comprehensive studies provide valuable insights into fungal diversity and evolutionary patterns. In a broader context, Lücking et al. [31] extensively discussed the significance of phylogenomic data in establishing species boundaries, particularly in lichen-forming fungi.

Nonetheless, an increasing number of fungal species and lineages are only known from their DNA sequences and cannot be associated with any physical specimens or established taxonomic names. These dark taxa fungi, also referred to as “unnamed and uncultured dark matter taxa”, represent the vast majority of fungal diversity [98]. This can be problematic, as they may be overlooked in legal and conservation efforts, as well as in counts of species diversity [99,100].

In recent decades, taxonomic studies of fungi have advanced significantly through the use of integrative (polyphasic) taxonomy, a method that combines genetic data (such as fungal DNA barcoding and phylogeny), physiological and biochemical features, ecological roles, and reproductive biology (when possible) to delimit species [31,101,102,103]. Recent advances in sequencing technology (metabarcoding, metagenomics, HTS, and WGS) have led to a greater understanding of fungal diversity, taxonomy, ecological roles, missing lineages, and fungal communities [64,104,105,106,107]. For example, metagenomics, specifically through metabarcoding and shotgun metagenomics, is instrumental in studying microbial communities. It allows for the analysis of fungal diversity in different environments, such as soil and aquatic ecosystems, as well as investigating the dynamics of microbial communities in food processing and fermentation [33,83,86]. Metagenomics also enables the exploration of the fungal microbiome response to diets, lifestyles, and environmental factors, providing insights into their roles in human health, pollution resilience, and ecological consequences of climate change [18,108]. Notably, shotgun metagenomics is a powerful and unbiased approach that offers a comprehensive view of microbial communities, enabling detailed analysis of their genetic composition, functional potential, and ecological roles across diverse environments [105,106,107].

Genomic data can also be used to understand how fungi have adapted to their specific lifestyles, such as the emergence of pathogens in the class Dothideomycetes and changes in microenvironments/microhabitats [109]. Moreover, phylogenomic studies have greatly improved our understanding of the tree of life and offered directions for future fungal phylogenetic and taxonomic studies [110]. Large datasets of fungal genomic and transcriptomic data have enabled the study of fungal evolution from a molecular sequence perspective in various fungal groups, i.e., the *Fusarium solani* species complex [111,112], microsporidia [113], asexual/sexual aspergilli [114], *Coprinopsis cinerea* and *Pleurotus pulmonarius* [115,115], arbuscular mycorrhizal *Paraglomus occultum* [116], and *Coccidioides immiti* [117]. WGS of diverse fungal taxa has revealed valuable information regarding large-scale genetic changes, including variations in chromosome structure and number that occurred during the evolution of fungi [109,110,111,112,113,114,115,116,117,118,119,120,121].

### 2.5. Fungi from Extreme Environments

Organisms that have adapted to thrive in extreme environments, such as deserts, the intertidal zone, oligotrophic seas, acid mine drainage, glaciers, the Arctic region, and hydrothermal vents, possess unique characteristics that make them desirable for industrial applications [122,123,124,125,126,127,128]. These organisms are adapted to growth under harsh conditions such as high salinity, high pressure, UV radiation, low oxygen concentration, hydrophobic conditions, high or low temperatures, acidic or alkaline pH, toxic compounds, and heavy metals. They are a promising source of various biomolecules for biotechnological, pharmaceutical, cosmetological, and industrial applications [129,130].

Several fungal microorganisms, including anaerobic fungi, lichen-forming fungi, true marine/marine-derived fungi, black fungi, halotolerant yeast, and *Aspergillus*, possess unique physiological and morphological adaptations that enable them to thrive in extreme conditions [131,132,133,134]. They are also known to produce a wide range of bioactive compounds such as enzymes, pigments, antibiotics, and other secondary metabolites that have potential applications in the pharmaceutical, cosmeceutical, and chemical industries [94,135,136]. The anaerobic fungi are of particular interest for their biotechnological applications and have been extensively studied using various omics approaches [131]. Integrated omics approaches can provide even greater insights, as shown by a study of the molecular mechanisms of degradation of polycyclic aromatic hydrocarbons by anaerobic fungi at both the single species and community levels [137]. By combining genomics, transcriptomics, proteomics, and other omics disciplines, researchers were able to obtain comprehensive insights into the intricate biochemical pathways and genetic elements driving the degradation of these environmental pollutants.

Anaerobic fungi, including the Neocallimastigomycota, have garnered significant attention due to their biotechnological applications, and researchers have extensively employed various omics approaches to study them [131]. However, to gain a deeper understanding of the complex molecular mechanisms involved in fungal processes, integrated omics approaches have emerged as powerful tools. This was exemplified by a recent study investigating the degradation of polycyclic aromatic hydrocarbons by anaerobic fungi, which employed integrated omics techniques to unravel the molecular mechanisms at both the single species and community levels [137]. By combining genomics, transcriptomics, proteomics, and other omics disciplines, researchers were able to obtain comprehensive insights into the intricate biochemical pathways and genetic elements driving the degradation of these environmental pollutants. In advancing our understanding of uncultivable fungi, high-throughput sequencing (HTS) cultivation-independent omics approaches have played a crucial role [108,138]. These techniques have been particularly instrumental in studying anaerobic fungi present in the rumen microbiome [139], including the elusive Neocallimastigomycota [140].

The Neocallimastigomycota are fungi commonly found in the digestive tracts of large herbivorous mammals and are known to produce natural products that have potential applications as antimicrobials, therapeutics, and other bioactive compounds [141]. The genomic information of these fungi was used to construct the first genome-scale metabolic model of an anaerobic fungus, which was experimentally validated and provided insights into the metabolic characteristics of gut fungi [142]. Omics-based technologies have enabled the characterization of the fungal population of anaerobic fungi in the animal rumen during plant cell wall hydrolysis [143], the identification of new promising enzymes [144], and the acquisition of a predictive comprehension of anaerobic communities for the purpose of directing microbiome engineering [145]. Microbiome engineering refers to the intentional alteration of microbial communities in specific environments, such as the human gut or soil, with the aim of attaining specific objectives. Its objective is to optimize the composition and functionality of the microbial communities to enhance human health, improve ecological processes, or optimize industrial applications. Probiotics, fecal microbiota transplantation (FMT), targeted interventions, and microbial manipulation are among the strategies employed in this field to achieve these goals [145].

Furthermore, the application of omics technologies, including glycomics (the study of carbohydrates and their biological functions), has played a pivotal role in enhancing our comprehension of fungi, their diversity, and their behavior in underexplored marine environments, specifically the deep sea (greater than 1000 m below sea level), which is considered as one of the most extreme environments of the ocean [20]. These technologies have proven to be a valuable source of bioactive molecules from marine fungi [132,133,134]. These techniques can be used to identify potential anti-infective drugs derived from extreme and unique environments [146]. Additionally, various aspects of the impact of genomics, transcriptomics, and proteomics have been studied in relation to black yeast [147] and have provided new directions for medical mycology such as using HTS-based approaches applied to mucormycosis caused by several fungal species belonging to the subphylum Mucoromycotina, order Mucorales, based on their genome structure, drug resistance, diagnostic development, and fungus–host interaction [148]. Furthermore, the integration of stable-isotope-enabled metabolomics with genomics, transcriptomics, and proteomics has been employed to pinpoint metabolites linked to the adaptation of fungi in acidic, metal-rich environments. This approach can offer valuable insights into the ecological role of fungi within communities [149]. Recent research has focused on understanding the physiology of the halophile *Aspergillus sydowii* as a model fungus for the study of molecular adaptations at saturated NaCl concentrations [150]. Additionally, omics studies have been used to investigate the characteristics of Aspergilli (*A. awamori*, *A. niger*, *A. oryzae*, *A. sojae*, and *A. terreus*) with respect to their tolerance of extreme cultivation conditions, their ability to grow on plant biomass, their high secretion capacities, and their versatile secondary metabolism. A greater understanding of these characteristics is crucial for developing these fungi as cell factories for producing organic acids, plant-polysaccharide-degrading enzymes, and secondary metabolites [151]. Omics approaches have been used to gain insights into how mycotoxigenic fungi adapt to environmental stresses and various interacting environmental conditions, and their relationship with phenotypic toxin production [152]. The advancement of metaomics approaches, which involve the thorough analysis of genetic material from entire microbial communities in specific environments, has been greatly improved by recent advances in high-throughput sequencing and bioinformatics. These advancements have significantly enhanced our capacity to assess fungal diversity, understand their potential functions, and monitor soil quality, thereby ensuring sustainable food production [153].

## 3. OMICS in Food-Related Fungi and the Food Industry

### 3.1. Food Safety and Security Based on Omics Techniques

Food safety and security are pressing concerns as the human population continues to grow. The concept of food safety and security encompasses four essential elements, namely availability, access, utilization, and stability. The limited availability of food has necessitated the adoption of various measures, including establishing organizations such as the World Food Program by the United Nations’ Food and Agriculture Organization, increasing agricultural productivity, providing agricultural insurance, forming global partnerships, and the large-scale storage of food. Nonetheless, numerous challenges still exist with respect to global food security, including climate change, water scarcity, agricultural diseases, fuel, land degradation, food sovereignty, and politics [154]. In addition, food safety is an essential aspect of food science, which focuses primarily on preventing foodborne pathogens (mostly bacteria and fungi [80,155,156,157] from contaminating food during all stages of food production, including harvesting, handling, and storage [158]. This poses a significant challenge to the food production process, requiring constant vigilance to prevent such occurrences.

### 3.2. The Role of Fungi in the Food Industry

The role of fungi in the food industry is vast, and they have numerous potential applications in both food and feed processing industries. Fungi produce various bioactive metabolites, pigments, colorants, antioxidants, oligosaccharides, and enzymes that are widely used in the food industry. Multiomics approaches have been employed to identify different kinds of fungal products and analyze their potential applications in food security [157]. Fungi are also consumed as processed foods, fodder, and fermented foods. Fungal biomass has been utilized to produce mycoproteins, which can be used as meat substitutes such as *Laetiporus sulphurous*, *Fusarium venenatum*, *F. oxysporum*, *Lentinula edodes*, *Aspergillus oryzae*, and *Fistulina hepatica.* Fungal-based white (industrial) biotechnology techniques are emerging as a significant contributor to food security [154]. However, fungi can also have adverse effects on food security, such as causing food spoilage, foodborne illnesses, toxins, and diseases, which can ultimately damage food production. Additionally, fungi can have a detrimental impact on global crop production and harvesting, including domestic animals [155,158].

To ensure food safety and security, it is imperative to enhance the detection of fungi and their metabolites. Achieving this is a significant challenge that requires novel strategies and biotechnological solutions. In response, many fields of research have transitioned from classical methodologies to advanced technologies in recent decades. These technologies have been used to improve food crops, reduce environmental impacts, and produce alternative sources of protein.

### 3.3. Macrofungi and Edible Mushrooms

Macrofungi, which include Basidiomycota and a few Ascomycetous members, are used as human food in the form of mushrooms, dietary supplements, and beverages of fungal origin [159]. Although there are around 14,000 species of macro fungi, only about 350 species are consumed as food, such as the widely cultivated *Agaricus bisporus*, *Lentinula edodes*, and *Flammulina velutipes* [160,161,162,163]. Mushrooms are valued as human food because of their unique flavors, nutritional content, and health-promoting characteristics. Global mushroom production increased 13.8-fold to 42.8 million tons from 1990 to 2020 [164]. The market for edible mushrooms is expected to be worth USD 72.5 billion by 2027 [165].

To address the growing demand for edible macrofungi, researchers can employ omics technology to investigate their cultivation, breeding, and production. WGS and RNA sequencing (RNA-Seq) are particularly valuable tools for producing transgenic edible mushrooms with desirable characteristics, such as high nutrient and pharmaceutical value, and resistance to abiotic stress conditions [55]. Proteomic studies are also necessary to understand amino acid and enzyme biosynthesis pathways, while metabolome sequencing technology can be used to analyze the metabolic pathways of substances in edible fungi, including active ingredients, undiscovered small molecules, and secondary metabolites with pharmaceutical effects, and to discover metabolomic markers to recognize edible macrofungi [55,166].

### 3.4. Foodomics

Omics approaches are crucial for food and nutritional security by providing not only targeted analyses of biomolecules but also a better understanding of biological processes at the system level [27,156,158,167]. The application of omics technologies in the food and nutritional domains is referred to as “foodomics” [27,167,168]. Foodomics enables a better understanding of how food safety and security can be maintained while meeting human health requirements, particularly by studying food contaminants and toxicity to ensure a secure food supply chain [27,158].

Notable technologies used in foodomics include WGS, pulsed-field gel electrophoresis (PFGE), multiple-locus variable-number tandem repeat analysis (MLVA), and RNA-Seq. WGS can identify fungal species present as food contaminants. PFGE and MLVA can monitor the spread of pathogens within a food processing plant. RNA-Seq can monitor transcript abundance patterns in food samples, which change during microbial colonization, providing valuable insights into the mechanisms involved in fungal contamination. Furthermore, foodomics approaches can be used to prepare food safety legislation for specific types of food associated with fungi and other microbes [27,158].

Fungal biotechnology also offers solutions to ensure food safety and security that can be a part of the circular economy, mainly due to improvements in fungal cell factories [169]. Overall, foodomics offers a promising strategy for detecting and managing fungal contamination in food, and the continued advancement of high-tech approaches will improve our ability to ensure food safety [27,158]. These examples demonstrate how multiomics approaches and tools can be used to address fungal threats to food safety and security. However, the use of these cutting-edge technologies can also present challenges. The large datasets generated by these omics approaches require careful data mining, reliable comparative analysis, and accurate statistical interpretation. Additionally, it is necessary to maintain comprehensive data banks and databases to store and manage the vast number of omics data.

The use of software tools has enabled access to omics datasets and an improved understanding of biological processes. However, the reliability of datasets needs to be constantly improved and upgraded. One challenge is obtaining adequate sample sizes and avoiding experimental design pitfalls to prevent overfitting and excessive false discoveries. It is important to address these issues to obtain real outputs and enable effective data sharing and mining [169,170]. Additionally, omics datasets are not static for both database providers and users, which presents another challenge that needs to be addressed [169].

### 3.5. Fungal Secondary Metabolites

Fungi produce a diverse range of secondary metabolites, including vitamins, amino acids, pigments, and antibiotics, which have numerous biotechnological applications, such as in agrochemicals, pharmaceuticals, agriculture, food, and cosmetic products. These metabolites have been found to possess anti-inflammatory, antioxidant, antimicrobial, and anticancer properties [171]. Fungal pigments are increasingly being used in the food industry due to their low production costs, easy processing, and consistent production yields [172]. These pigments are also safer for human health and the environment compared with synthetic pigments [173] (Table 1). Despite the potential use of fungal pigments as food additives, their usage is restricted because of the potential presence of naturally occurring toxic secondary metabolites (mycotoxins). One such example is *Monascus*, which is a promising source of natural colorant, but is prohibited in the European Union and the United States due to the presence of the mycotoxin citrinin [174].

Fungal secondary metabolites are important sources of food flavors. Vanillin, for example, is primarily produced by engineered microorganisms rather than vanilla plants, with *Aspergillus niger* and *Pycnoporus cinnabarinus* being used to produce it from waste residues of rice bran oil, whereas *Schizosaccharomyces pombe* and *Saccharomyces cerevisiae* are used to produce it from inexpensive glucose. Benzaldehyde, which is an important flavoring compound for baked goods, can be produced using phenylalanine by *P. cinnabarinus*. In addition, *S. cerevisiae* has been used to reconstruct the primary aroma compound in raspberries, [4-(4-Hydroxyphenyl) butan-2-one] [175,176,177,178,179].

Plants and microorganisms are the main sources of natural metabolites. The cultivation of fungi is not affected by seasonal or geographical variations like plant crops, and fungi can be genetically engineered for increased metabolite production. The advantages of fungi over plants also include high growth rates, small space requirements, and the ability to be cultivated in inexpensive media with high biomass concentrations [180]. Despite the advantages of metabolite production from fungi, the potential of fungi to produce secondary metabolites for industrial application has not been fully realized yet, as most gene clusters responsible for secondary metabolite biosynthesis are only expressed under stress conditions and are silent under standard cultivation conditions [181]. To expand the potential pool of secondary metabolites, various approaches such as multiomics analyses, gene cluster activation, chemical genomics, metabolic identification, and genetic engineering can be utilized [171].

It is essential to explore alternative food sources, including alternative protein sources, as a means of reducing food security risks. Fungal enzymes have a crucial role in the food industry, as demonstrated by the use of amylases from *A. niger* and *A. oryzae*, proteases from *A. oryzae*, pectic enzymes from various *Aspergillus* species, galactosidase from *Mortierella vinaceae*, lactase from *A. oryzae* and *A. niger*, and invertase from *Saccharomyces* species [182,183,184]. *Aspergillus oryzae* is used for fermenting traditional Japanese foods like sake, shoyu, miso, and vinegar. Fungi also produce important vitamins used in the food industry, such as vitamin B2 (riboflavin), which is synthesized by *Candida guilliermondii, Debaryomyces subglobosus*, and *Ashbya gossypii* [185]. *Mortierella alpiney* has the capability to synthesize longer polyunsaturated fatty acids [186]. The nutraceutical properties of edible fungi like *Lentinula edodes*, *Ganoderma lucidum*, *Tremella mesnterica*, *Hericium erinaceus*, *Sclerotinia sclerotiorum*, *Cordyceps sinensis*, and *Trametes versicolor* are responsible for their popularity [187]. Since 1985, mycoprotein extracted from *Fusarium venenatum* has been used as a food-grade protein source with a texture similar to meat that can be frozen, canned, and dried. Mycoproteins are versatile and can be combined with different food items such as biscuits, soups, and fortified drinks. Analyzing the genetic makeup and nutritional value of these alternative food sources can be achieved through a combination of genomics, proteomics, and metabolomics [188].

### 3.6. Mycotoxins and Fungi

Fungi are a major cause of damage to cereal production worldwide, affecting major crops such as wheat, maize, and rice. Therefore, the detection of fungi and their metabolites is important in the food industry to ensure food security [155,189]. Fungal diseases also pose a threat to other species, such as *Pseudogymnoascus destructans*, which can cause catastrophic epidemics in bats and a concomitant increase in crop-destroying insects in fields [155]. Mycotoxins can contaminate a range of food products including meat, milk, eggs, and field crops [190]. *Aspergillus* and *Penicillium* are common mycotoxin-producing fungi that can contaminate food products [156]. Mycotoxins can be detected using metabolomic approaches [156,158,190].

The metabolomic approach has been successful in detecting various types of mycotoxins produced by different fungal species, including *Alternaria, Fusarium*, and *Claviceps*, which have the highest toxigenic potential [190]. These mycotoxins include Citrinin, Aflatoxins, Fumonisins, Zearalenone, Ochratoxins, Ergot Alkaloids, Patulin, Tremorgenic toxins, and Trichothecenes, which can be found in various foods [156,158,191]. The main fungal genera that are represented among food pathogens and mycotoxins in food industries that are the focus of foodomics applications include: (a) *Aspergillus* responsible for Aculeacin A, B, C, D, E, F, and G; Aflatoxin B; Aflatrem; and Ochratoxin; (b) *Penicillium* responsible for Citrinin, Amauromine, Agroclavine, and Patulin; (c) *Claviceps* responsible for Aflatrem, Chanoclavine I, Ergochromes, Ergobutine, Ergobutyrine, and Ergobine; and (d) *Fusarium* responsible for Deoxynivalenol, Fumonisins, Trichothecenes, and Zearalenone [156]. Metabolomics has provided insights into the interactions between phytopathogenic fungi and their hosts. Metabolomic studies of phytopathogenic fungi including *Rhizoctonia solani*, *Botrytis cinerea*, *Ustilago maydis*, *Sclerotinia sclerotiorum*, *Magnaporthe oryzae*, and *Fusarium graminearum* have revealed mechanisms of fungal infection and plant defense [192]. The interactions between fungal pathogens and plants are vital for global agricultural production and food security and have been widely researched [192,193]. Besides the study of mycotoxin-producing phytopathogenic fungi, metabolomics is useful for identifying fungal endophytes that produce bioactive compounds in a host-dependent manner [194]. In addition to the metabolomic study of interactions between phytopathogenic fungi and hosts, the impact of interacting fungi on the mineral and elemental composition of plants can be revealed by ionomics [167,195], notably, the effect of arbuscular mycorrhizal fungi inoculation on the growth of maize under various environmental stressors [196].

Fungal omics approaches such as transcriptomics and proteomics are important for the development of biomarkers and biosensors of mycotoxin-producing fungi [170]. *Aspergillus flavus* produces aflatoxin, a major contaminant of several crops including groundnut and maize. Transcriptomic and proteomics approaches have been used to identify genes and proteins associated with resistance to aflatoxin contamination in groundnut and maize, leading to understanding the host defense mechanism that includes pathogenesis and antioxidant-related genes involved in the suppression of aflatoxin biosynthesis or its detoxification [197]. The study of phosphorylated proteins (phosphoproteomics) has also revealed the ability of crops such as wheat and grapevine (*Vitis vinifera*) to resist a fungal pathogen (*Septoria tritici*) [198].

The knowledge of candidate biomarkers of resistance to fungal pathogens obtained from omics approaches has spurred efforts to create pathogen-resistant transgenic crop varieties with modifications of resistance-associated genes. For example, transgenic finger millet crops have been developed with enhanced gene expression to combat fungal blast disease and improve yield [199]. The generation of transgenic plants has benefitted from the emergence of genome editing technology. Clustered regularly interspaced short palindromic repeats (CRISPR) and CRISPR-associated (Cas) protein genome editing tools offer a cost-effective and versatile approach to generate transgenic plants with modifications of genes associated with traits of interest. Genome-editing of plants has been used to generate high-yielding and stress- and disease-resistant crop varieties [200,201,202]. Of particular interest, genome editing technology has been used to knock out genes associated with susceptibility to fungal pathogens, including the rice blast pathogen *Magnaporthe oryzae* and the powdery mildew pathogen *Podosphaera xanthii* [201].

### 3.7. Food Industry

Among food industries, the dairy industry is the most impacted by fungi. Spoilage of dairy products by molds poses a major food safety challenge. To control molds, antifungal lactobacilli species like *Lacticaseibacillus rhamnosus* and *L. paracasei* can delay spoilage and increase the shelf life of dairy products. Metabolomics can identify the key compounds that are essential for antifungal activity. This approach has been successfully applied against *Penicillium commune* and *Mucor racemosus*, resulting in the development of new protective strains [203]. Additionally, meta-transcriptomics (RNA-Seq of complex community microbial samples) has led to insights into the role of fungal microflora such as *Geotrichum candidum* and *Penicillium camemberti* in the cheese ripening process [204]. The use of omics approaches and their applications has proven to be efficient in producing safe foods and ensuring food security (Figure 1). Furthermore, these applications are effective and productive tools for conducting systems biology investigations and studying fungi [27,195].

### 3.8. Postharvest Losses

Postharvest losses of food affect quality, nutrition, seed viability, and market value [205]. The global postharvest food loss has been calculated to be approximately 1.3 billion tons annually, which disproportionally affects developing countries. For instance, post-harvest losses account for 30–40% of fruits and vegetables produced in India [206]. A major cause of postharvest loss of fresh fruits and vegetables is pathogenic fungi. *Penicillium* spp., *Botrytis cinerea*, *Alternaria alternata*, *Monilinia* spp., *Trichothecium roseum*, *Fusarium* spp., and *Colletotrichum* spp. are responsible for the majority of postharvest losses [207].

Understanding the infection process mechanism of fungal pathogens is crucial for mitigation of post-harvest diseases. Phytotoxic metabolites, secreted proteins, and small RNAs of fungal pathogens contribute to the infection process. At the early stage of infection, necrotrophic pathogenic fungi kill host cells and develop necrotic areas for successful colonization [207]. Proteins secreted by the pathogenic fungus *Fusarium proliferatum* in the infection process of banana peel were identified by comparative proteomics [78], and cell-wall-degrading enzymes and secondary metabolites secreted by the pathogenic fungus *Monilinia fructicola* were identified by sequence analyses and gene expression studies [208]. Furthermore, analysis of *B. cinerea* mutants identified genes encoding the cell-wall-degrading enzymes cellobiohydrolase and xylanase to be essential for virulence [209].

When a pathogenic fungus attacks a plant, reactive oxygen species (ROS) accumulate around the infection site as part of the plant defense mechanism [210]. ROS derived from pathogenic fungi also play a significant role in the infection process [211]. In fungi, the NADPH oxidase complex (Nox) is the most important enzyme complex for ROS production. The reduction in vegetative growth, conidia formation, and loss of virulence in *B. cinerea* were observed by *NoxR* gene knockout [79]. In addition to ROS, small non-coding RNAs (sRNAs) play roles in regulating plant immunity against pathogen infections [212]. The pathogenic fungus *B. cinerea* produces sRNAs that hijack the host RNA interference machinery and selectively silence host plant immune genes [213]. In addition to sRNAs, pathogen-protein-coding genes important for virulence have been identified using the gene knockout approach. Using this approach, the MAP kinase genes *Pdos2, PdSlt2,* and *PdMpkB* in the signal transduction pathway were shown to regulate the pathogenicity of *Penicillium digitatum* [214], and transcription factors regulating development and pathogenicity were identified in *Fusarium graminearum* [215].

## 4. Biomedical and Therapeutics Applications Based on the Omics Techniques

Despite advances in diagnostic and treatment methods, the incidence of invasive fungal infections in humans is rising rapidly. Since 2020, disruptions in public health measures owing to the COVID-19 pandemic have led to a rise in invasive fungal infections, with Aspergillosis, mucormycosis, and candidaemia being common fungal infections with fatal consequences. In response to this threat, the World Health Organization (WHO) released the fungal pathogens priority list (FPPL) in October 2022. This list includes 19 fungi that pose the highest health risk and is the first global effort to prioritize fungal diseases based on research and development (R&D) needs and public health implications [216].

The WHO FPPL has categorized 19 fungal pathogens into three priority groups based on their public health implications and R&D needs [216]. The priority group includes *Aspergillus fumigatus*, *Cryptococcus neoformans*, *Candida auris*, and *Candida albicans*. The high-priority group includes *Nakaseomyces glabrata* (=*Candida glabrata*), *Histoplasma* spp., eumycetoma causative agents, Mucorales, *Fusarium* spp., *Candida tropicalis*, and *C. parapsilosis*. The medium-priority group includes *Scedosporium* spp., *Lomentospora prolificans*, *Coccidioides* spp., *Pichia kudriavzeveii* (=*Candida krusei*), *Cryptococcus gattii*, *Talaromyces marneffei*, *Pneumocystis jirovecii*, and *Paracoccidioides* spp. The aim of the WHO FPPL is to prioritize research and policy measures to improve global responses to fungal infections and antifungal resistance [216]. Currently, there are only four categories of antifungal drugs (pyrimidines, azoles, polyenes, and echinocandins) used in clinical settings. These drugs have been found to be efficacious in treating fungal infections, but they can also have negative impacts on patients’ recovery and outlook, as indicated by multiple studies [216,217,218,219,220]. 

### 4.1. Antifungal Drugs

Due to the conservation of essential eukaryotic genes between fungi and humans, there are a limited number of targets for developing safe and effective antifungal drugs [169,221]. Most antifungal medications work by disrupting the synthesis or integrity of ergosterol, which is the primary sterol found in fungal, but not human, cell membranes. Some drugs cause breakdown of the fungal cell wall. Flucytosine is a pyrimidine analog that prevents the production of nucleic acids and proteins by fungal cells, thereby inhibiting their growth. It is effective in treating systemic infections caused by *Candida* and *Cryptococcus* species [221,222]. Azoles are the most commonly used antifungal medications that inhibit the lanosterol 14α-demethylase enzyme (encoded by the ERG11 gene) in the fungal cytochrome P450 pathway. There are two main groups of azoles, namely imidazoles (ketoconazole, miconazole, clotrimazole, and econazole) and triazoles (fluconazole, itraconazole, voriconazole, and posaconazole) [221,222].

The *Streptomyces* comprises polyene-producing taxa, which have broad-spectrum activity against various fungal species. These drugs work by binding to ergosterol in the fungal cell membrane, leading to the development of transmembrane holes, depolarization of the membrane, and ultimately, cell death. Echinocandins are cyclic lipopeptides with a semisynthetic acyl lipid side chain. Caspofungin, micafungin, and anidulafungin are examples of echinocandins that inhibit (1,3)-D-glucan synthase, a crucial component of fungal cell walls that is not present in humans. The lack of (1,3)-D-glucan production in fungi leads to osmotic instability and eventual cell death [221,222,223]. Figure 2 illustrates the targets of current antifungal drugs, and the drug generic names.

### 4.2. Molecular Mechanisms of Antifungal Drug Resistance

The surge in the number of fungal infections has led to a rise in research for new antifungal treatments. Although current commercially available antifungal drugs primarily aim at the cell wall and plasma membrane, alternative targets have been identified. Recent studies have concentrated on identifying new drugs targeting fungal virulence factors [224].

Antifungal drug resistance has become a concern due to the limited number of available options to treat the increasing number of opportunistic fungal infections. Azole resistance is primarily attributed to overexpression of the ERG11 drug target gene [225] and multidrug efflux pumps, including ATP-binding cassette (ABC) superfamily and major facilitator superfamily proteins [224,226,227]. The uptake of flucytosine involves cytosine permease, which is then converted to 5-fluorouracil by cytosine deaminase. Mutations in the cytosine deaminase gene confer resistance to Flucytosine [228,229]. Echinocandin resistance has been attributed to mutations in the 1,3-beta-D-glucan synthase FKS1 target gene that reduce the binding affinity for these drugs [230]. Changes in enzymes involved in ergosterol synthesis are the primary cause of polyene resistance. Mutations in the sterol C-5 desaturase ERG3 gene in *Candida* are associated with reduced ergosterol levels in the fungal membrane and resistance to azoles and polyenes [231,232]. In addition to these mechanisms, the formation of biofilms by fungi has also been identified as a leading factor contributing to multidrug resistance in fungi [233,234].

### 4.3. Fungal Omics Data for Novel Drug Development

In the past two decades, mycology research has been transformed by the use of mass-spectrometry-based proteomics, which allows for the measurement of protein synthesis, posttranslational modifications, alternative protein isoforms, and interaction networks. A number of studies on fungal omics data are available, and we recommend that readers consult them for a comprehensive understanding of the utility of fungal omics data [25,169,235,236,237,238]. In this section, we focus on two types of fungal proteins that have the potential to become important targets for future drug development.

Multidrug resistance is often linked to the increased expression of pleiotropic drug resistance (PDR) efflux pumps belonging to the ABC transporter superfamily [239]. PDRs have a distinctive structure compared with other well-studied types of ABC proteins. Both prokaryotes and eukaryotes have membrane-bound ABC proteins that transport substrates across organelles and cell borders, enabling nutrients to enter cells and harmful substances to exit them. In general, eukaryotic ABC proteins are made up of two identical halves, each containing a nucleotide-binding domain (NBD) and a transmembrane domain (TMD) with six transmembrane spans (TMSs) with either extracellular (EL) or intracellular (IL) loops between adjacent TMSs [239,240]. Unlike human ABC proteins, fungal ABC proteins have a unique domain arrangement of [NBD-TMD]2, with distinctive ELs containing amino acid residues conserved in fungi that are absent from human ABC pumps [241]. Targeting these ELs with drugs may provide a promising avenue for future antifungal drug development, as they are surface-accessible and not subjected to cellular efflux or detoxification pathways [239,240,241,242].

When pathogens infect hosts, they face various stresses such as heat shock and oxidative stress, and their survival relies on their ability to respond to these external conditions at the molecular level. The stress response in the context of host–pathogen interactions has recently been identified as an important and innovative mechanism for developing antimicrobial agents [243]. However, developing inhibitors that target the stress-response machinery of pathogens has not yet been successful, mainly due to the high-sequence conservation of heat-shock proteins (Hsps) across different domains of life [243].

The Hsp100 protein family includes promising targets for developing novel antimicrobial inhibitors. Hsp100 chaperones, which rely on ATP, are essential for the survival of lower eukaryotes, plants, and bacteria (where they are referred to as ClpB) under stress conditions. In microbial cells that experience stress, Hsp100s play a role in exclusively disentangling misfolded proteins. Unlike other heat-shock protein families, Hsp100s are not present in animals or humans [243,244,245]. Hsp100 chaperones play a crucial role in the invasiveness and survival of several important protozoan and bacterial pathogens, including the ESKAPE pathogens (*Enterococcus faecium*, *Staphylococcus aureus*, *Klebsiella pneumoniae*, *Acinetobacter baumannii*, *Pseudomonas aeruginosa*, and *Enterobacter* spp.) responsible for the majority of nosocomial infections [244,245,246]. In yeast, these chaperones are referred to as Hsp104 proteins, which are hexameric AAA+ proteins with an asymmetric ring-shaped translocase structure that are absent in metazoans [246,247].

Recently, Zokiewski and colleagues have proposed that inhibitors of AAA + ATPases, distantly related to Hsp100, may act as prototype scaffolds for developing Hsp100-selective ligands [244]. Although sharing only 45% of amino acid sequence identity, yeast Hsp104 and bacterial ClpB exhibit similar overall structures (Figure 3). On the other hand, the structural diversity of AAA + ATPases is sufficient to distinguish between ligands of the same chemical family [248]. This suggests that derivatives of antimicrobial drugs could be synthesized or be more selective for Hsp104. While molecular chaperones have not been explored as potential targets for novel antimicrobials, this approach could hasten the discovery of antifungal compounds and new drugs.

## 5. Conclusions and Future Prospects

This article highlights the importance of omics technologies in identifying the impact of fungi on different ecosystems and their multifaceted roles, which can be beneficial or detrimental. The limitations of the traditional method of identifying fungal species based on morphological traits are discussed, and advanced technologies such as PCR and DNA sequencing are presented as more accurate classification and identification tools. HTS and metagenomics of environmental samples are also highlighted as solutions to identify novel lineages in the Kingdom of Fungi. The analysis of protein-coding genes and whole genomes (phylogenomics) is revolutionizing our understanding of phylogenetic relationships and fungal taxonomy. The applications of metabolomics proteomics, transcriptomics, and metatranscriptomics are important for a comprehensive understanding of fungal metabolism. Advanced omics technologies are crucial in identifying and classifying diverse fungal species and broadening the understanding of fungi, which has significant implications for food safety and security, edible mushrooms, fungal secondary metabolites, and biomedical and therapeutic applications. Despite their benefits, individual omics technologies have certain limitations. For example, genomics can only provide information about metabolic potential, whereas transcriptomics and proteomics are often characterized by poor reproducibility [249,250,251]. The complexity of fungal genomes, proteomes, transcriptomes, and metabolomes presents a limitation in fully annotating and interpreting the functions of individual genes and molecules [122]. Biased sampling, low statistical power, and inappropriate methodology also pose challenges in omics studies [252,253]. Furthermore, the lack of genetic tools and resources, the high cost, the computational power and infrastructure, and the technical expertise required for omics studies can act as barriers [253,254]. Although multiomics approaches can offer a more comprehensive and precise understanding of fungal communities, there is still considerable room for improvement. Therefore, the integration of DNA and RNA sequencing and proteomic data in multiomics analysis is necessary to advance our understanding of fungi.

The future prospects of fungal omics are promising, with advancements in sequencing technologies, the integration of multiomics data, computational analysis, and bioinformatics tools. These developments will enable comprehensive exploration of fungal genomes and functional elements, leading to the discovery of novel species, pathways, and bioactive compounds. Fungal omics will continue to contribute to bioremediation, industrial biotechnology, agriculture, and medicine, while also providing insights into fungal–host interactions and personalized approaches for disease management. Continued collaboration, technological advancements, and interdisciplinary research will drive further advancements in fungal omics, paving the way for new discoveries and applications.

## Figures and Tables

**Figure 1 jof-09-00688-f001:**
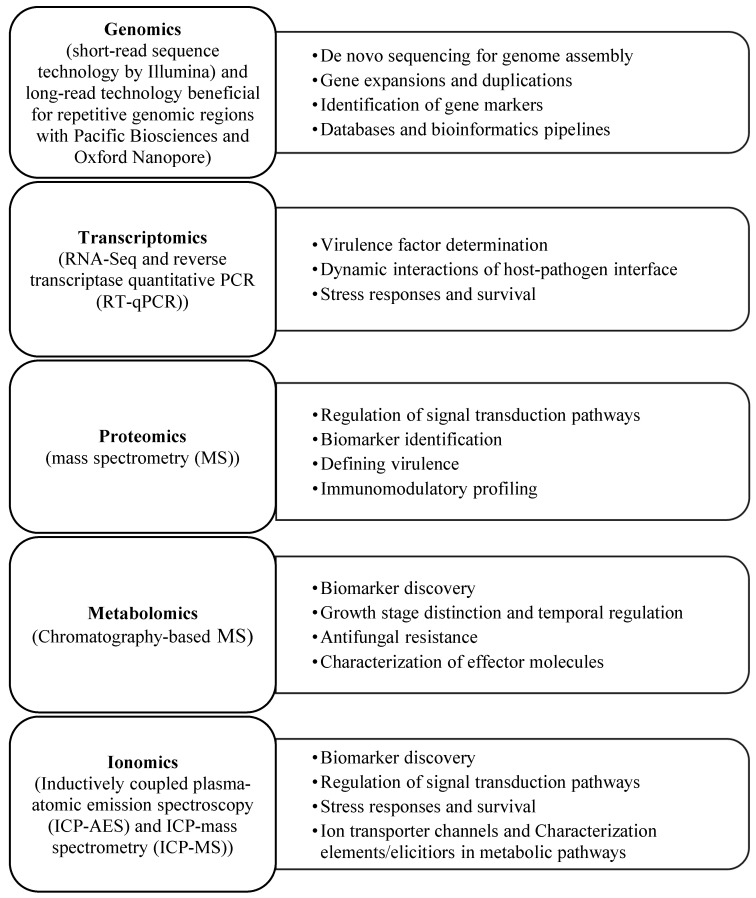
Outline of omics tools and applications in fungal omics to ensure food safety and security.

**Figure 2 jof-09-00688-f002:**
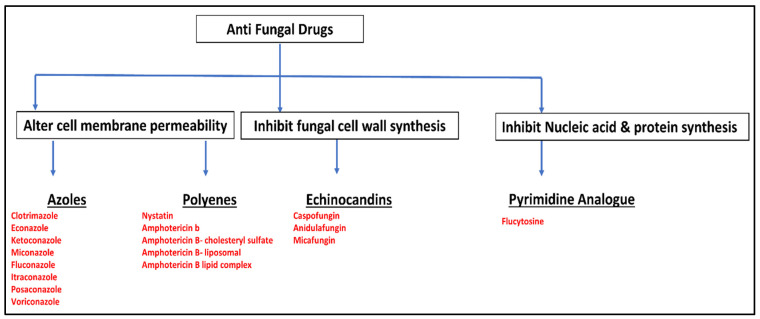
Targets of current antifungal drugs (boxed) and generic drug names (highlighted in red). Drug classes are underlined.

**Figure 3 jof-09-00688-f003:**
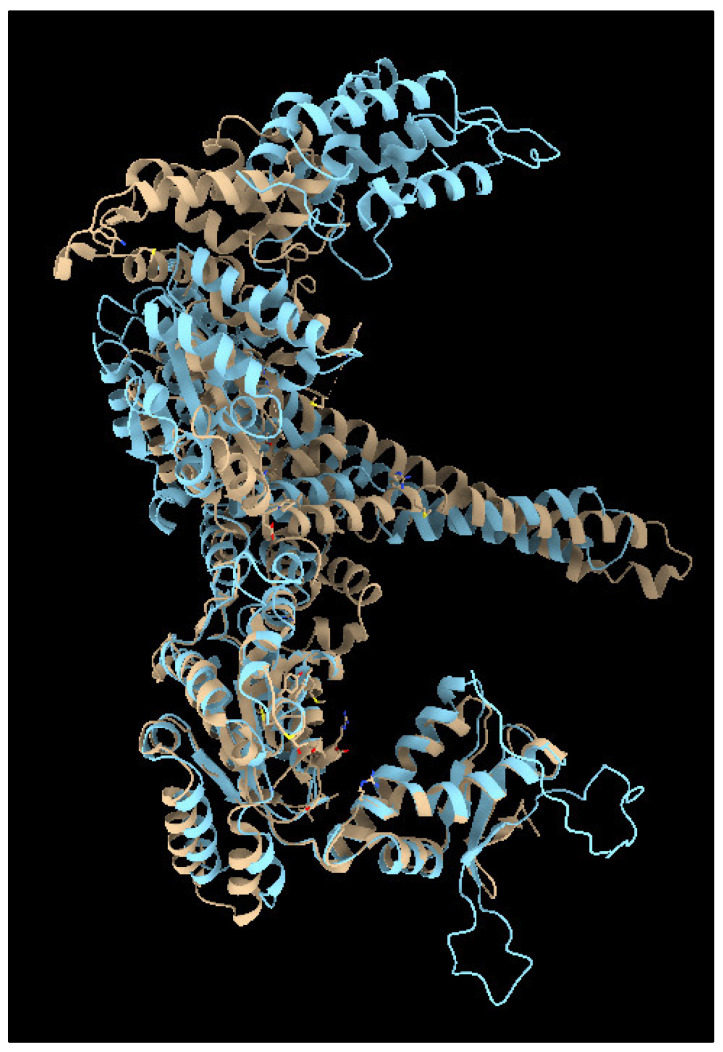
The monomeric structures of *E. coli* ClpB (depicted in light brown, PDB 1QVR) and *S. cerevisiae* Hsp104 (predicted by Alphafold as AF-P31539-F1, shown in light blue). Molecular graphics and analyses were performed with UCSF Chimera, by the Resource for Biocomputing, Visualization, and Informatics at the University of California, San Francisco.

**Table 1 jof-09-00688-t001:** Safety evaluation of fungal pigments used in the food industry.

Fungal Name	Pigment	Color
*Penicillium purpurogenum*	Mitorubrino Mitorubrin Purpurogenone Azaphilone	Orange–red Yellow Yellow–orange
*Rhodotorula glutinis*	Torulene β-Carotene Torularhodin	Red and orange
*Thermomyces* sp.	Naphthoquinone	Yellow
*Yarrowia lipolytica*	β-Carotene	Orange

## Data Availability

Data sharing is not applicable, and no new data were created or analyzed in this study. Data sharing is not applicable to this article.

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
