# Peer review of "OMICS and Other Advanced Technologies in Mycological Applications"

_jof, 2023, doi:10.3390/jof9060688_

Round 1
Reviewer 1 Report
Wijayawardene et al. give an interesting insight into the OMICS and other advanced technologies that have been applied in the field of mycology. The area has been thoroughly researched and well documented. However, there are certain points that need to be addressed.
The title “OMICS and other advanced Technologies Applied in Mycological Aspects” could be better framed as “OMICS and other advanced Technologies Applied in Mycology”
Abstract “both positive and negative impacts” impacts on whom or what?
Line 38 “fungal applications” for the advancement of mankind?
Line 71-72: “To delimitate species boundaries and un- 71 derstanding the metabolic pathways of these species is essential in utilizing them effec- 72 tively, thus, advanced technologies are being employed” rephrase.
Line 88, 92: “Mahato et al. [14]”; “to Schoch et al. in 2009 [27]” etc. are these in the right format? Check.
Line 180-181: “MALDI-TOF MS is used by various fungal groups for identification purposes” rephrase.
Line 199-202 “In addition, fungi have a wide range of biotechnological and pharmaceutical applications, including several bioactive 200 metabolites, pigments, dyes, antioxidants, polysaccharides, novel medicines, and industrial enzymes” here the authors can introduce, the terminology “ecosystem services” provided by the fungi.
A timeline (illustrated) of when these techniques mentioned (in the manuscript) were introduced into fungal research would be helpful, this gives an insight into technological advances at a glance.
Authors should also conclude or highlight the applications of other emerging technologies that have been used in other fields and have the potential to be applied in the field of mycology. The authors should also address the gaps, these types of approaches add more depth to their review and can avoid the documentation, and the writers can bring their own perspective to blend with the current writing style.
The language is fine, some minor corrections are required, which I have asked to rephrase.
Author Response
Response to Reviewer 1 Comments
Wijayawardene et al. give an interesting insight into the OMICS and other advanced technologies that have been applied in the field of mycology. The area has been thoroughly researched and well documented. However, there are certain points that need to be addressed.
Response 1: Thank you so much for overview suggestion, we have already recorrected as suggestion as follows:
The title “OMICS and other advanced Technologies Applied in Mycological Aspects” could be better framed as “OMICS and other advanced Technologies Applied in Mycology”
Response 2: We modified the title as suggested and changed it into “OMICS and Other Advanced Technologies in Mycological Applications”
Abstract “both positive and negative impacts” impacts on whom or what?
Response 3: We have re-modified the abstract
Line 38 “fungal applications” for the advancement of mankind?
Response 4: We modified as suggested
Line 71-72: “To delimitate species boundaries and understanding the metabolic pathways of these species is essential in utilizing them effectively, thus, advanced technologies are being employed” rephrase.
Response 5: We checked and rephrased
Line 88, 92: “Mahato et al. [14]”; “to Schoch et al. in 2009 [27]” etc. are these in the right format? Check.
Response 6: We checked and recorrected as the right format
Line 180-181: “MALDI-TOF MS is used by various fungal groups for identification purposes” rephrase.
Response 7: We checked and rephrased
Line 199-202 “In addition, fungi have a wide range of biotechnological and pharmaceutical applications, including several bioactive 200 metabolites, pigments, dyes, antioxidants, polysaccharides, novel medicines, and industrial enzymes” here the authors can introduce, the terminology “ecosystem services” provided by the fungi.
Response 8: We checked and rephrased
A timeline (illustrated) of when these techniques mentioned (in the manuscript) were introduced into fungal research would be helpful, this gives an insight into technological advances at a glance.
Response 9: Apologizing for this, we can’t’ provide
Authors should also conclude or highlight the applications of other emerging technologies that have been used in other fields and have the potential to be applied in the field of mycology.
Response 10: We gave more detail on highlight the applications of other emerging technologies in the field of mycology
The authors should also address the gaps, these types of approaches add more depth to their review and can avoid the documentation, and the writers can bring their own perspective to blend with the current writing style.
Response 11: We gave more detail on the gaps and our own perspective to blend with the current writing style.
Comments on the Quality of English Language
The language is fine, some minor corrections are required, which I have asked to rephrase.
Response 12: We sent the ms for the English Language’s service by English native speaker (AJE; https://www.aje.com) again for make sure all of texts should be corrected.
The revised MS with "Author to respond reviewer 1" is attached.

Reviewer 2 Report
This "Omics" focused review is solid, reasonably well organized and thorough. It should provide readers a solid introduction to the overall understanding of how "Omics" can be put to use in fungal studies. As a part of this special issue, it seems fitting and I have no concerns about any of the ideas presented or summarized. There are a few areas in need of tighter editing (a few missed words and duplicated sentences or sections) but the copy editors should catch these with close attention.
It is fine.
Author Response
Response to Reviewer 2 Comments
Comments and Suggestions for Authors
This "Omics" focused review is solid, reasonably well organized and thorough. It should provide readers a solid introduction to the overall understanding of how "Omics" can be put to use in fungal studies. As a part of this special issue, it seems fitting and I have no concerns about any of the ideas presented or summarized. There are a few areas in need of tighter editing (a few missed words and duplicated sentences or sections) but the copy editors should catch these with close attention.
Comments on the Quality of English Language
It is fine.
Response 1: Thank you so much for positive feedback. In case of the tighter editing in English (a few missed words and duplicated sentences or sections, we sent the ms for the English Language’s service by English native speaker (AJE; https://www.aje.com) again for make sure all of texts should be corrected.
The revised MS with "Response to Reviewer 2 Comments" is attached.

Reviewer 3 Report
Dear Authors!
Thank you for this interesting and potentially important contribution. I think this work can be of interest to the general mycological community, and also for people generally interested in various omics approaches.
However, at this stage the manuscript requires extensive editing and complete restructuring. As it is now, it almost seems like each of the chapters was written by the different author - each chapter has its own introduction, and often repeats the information already given. There is also a wealth of various information which seem to be given in the manuscript almost without a clear purpose. This really obscures the actual aims of this review and creates a significant chaos.
Your manuscript is entitled "OMICS and other advanced Technologies Applied in Mycological Aspects", and yet you start the Abstract with the sentence "This scientific review delves into the many roles of fungi in different ecosystems, including both positive and negative impacts." That would imply that you are rather focused on the ecology/biology of fungi, but the title and the majority of the content suggest obviously otherwise. You should therefore rewrite the Abstract, providing clear aims, and some clear, most important information.
As I understand, this is the review of recent advances in mycology using various "omics" approaches, and this should be your main goal - to explain what are those "omics" approaches, what are the potential applications, and what are the major recent breakthroughs achieved thanks to the use of these "omics" techniques. I think all that information is already there in your manuscript, and you only need to present clear aims, and then follow with a clear structure.
You have chosen to structure this review according to some topics (taxonomy and classification, food and industry, biomedical sciences), so maybe in the introduction you could for example introduce the overall generalised scheme of "omics" - maybe even slightly modified Figure 1 (and then obviously delete it from the section where it currently appears), then explain what are the "omics", and then only shortly introduce the importance of fungi and the reason for choosing this particular selection of topics? And then within every chapter, try to focus on clearly showing which of the "omics" methods had led to what breakthroughs. As it is now, there is very difficult to see the connection between different information pieces.
Some specific comments:
a) Lines 80-82 - here you mention the estimates of the fungal diversity, for the first time, already giving some numbers. But then, in lines 192-197 you repeat this information, giving actually additional estimates. I would just leave this information (the more detailed one) in only one place.
b) While I appreciate some in-depth information provided (e.g. lines 770-816 on antifungal drugs, together with the Figure 2), I feel that this is too much deviating from the fundamental aim of the paper which is the "recent advances of omics". I understand that you have to introduce some of these topics, but I would suggest to try to shorten them, so that they would not dominate the text.
c) Similar to the previous comment - lines 821-822 "A number of studies on fungal omics data are available, and we recommend that readers consult them for a comprehensive understanding of the utility of fungal omics data [25,70,176,249–251]. In this section, we focus on two types of fungal proteins that have the potential to become important targets for future drug development." - this is extremely confusing - so now you write that this paper is actually not about the "omics", but it will focus on some important proteins from now on... And you even provide the extensive description (lines 825-872) and the figure 3 with the structure of the protein. I do think this is interesting, I do think this is important, but suddenly, instead of clear review showing the application of the "omics", the Reader has a minireview on these proteins! I feel that this has to be contextualised in light of the "omics" methods used to make this discovery. There are several similar issues throughout the manuscript, where you delve too much, in my opinion, into description of the problem, without clearly referring to the role of the "omics".
d) Please carefully review the information you are giving, and provide citations (including separate citations) for important, information-bearing statements. Some smaller examples:
- Line 60 "Ellis and Sutton [4,5].", should read "...Ellis [4], and Sutton [5]."
- Line 79 "Mitchell and Zuccaro (2006) and Seifert (2009) [10,11]." - so what's the conception here? Shouldn't it be "Mitchell and Zuccaro [10], and Seifert [11]"?
d) You have the paragraph entitled "2.1. Discovering New Taxa in Known Lineages" (line 135), and yet the introductory lines 136-160 are focused on something totally different!
e) Please be consistent with omics and "omics" - both versions appear throughout the manuscript.
I really feel that you need some English correction there, and this is the work for English-editing company.
This concerns mostly grammar, but also very often the logical meaning of the sentence.
Author Response
Response to Reviewer 3 Comments
Thank you for this interesting and potentially important contribution. I think this work can be of interest to the general mycological community, and also for people generally interested in various omics approaches.
Response 1: Thank you so much for positive feedback.
However, at this stage the manuscript requires extensive editing and complete restructuring. As it is now, it almost seems like each of the chapters was written by the different author
Response 2: we have already recorrected as suggestion
- each chapter has its own introduction, and often repeats the information already given. There is also a wealth of various information which seem to be given in the manuscript almost without a clear purpose. This really obscures the actual aims of this review and creates a significant chaos.
Response 3: we have already recorrected as suggestion
Your manuscript is entitled "OMICS and other advanced Technologies Applied in Mycological Aspects", and yet you start the Abstract with the sentence "This scientific review delves into the many roles of fungi in different ecosystems, including both positive and negative impacts." That would imply that you are rather focused on the ecology/biology of fungi, but the title and the majority of the content suggest obviously otherwise. You should therefore rewrite the Abstract, providing clear aims, and some clear, most important information.
Response 4: we re-write the Abstract, for providing clear aims, and some clear, most important information as suggested
As I understand, this is the review of recent advances in mycology using various "omics" approaches, and this should be your main goal - to explain what are those "omics" approaches, what are the potential applications, and what are the major recent breakthroughs achieved thanks to the use of these "omics" techniques. I think all that information is already there in your manuscript, and you only need to present clear aims, and then follow with a clear structure.
Response 5: we have already recorrected based on the potential applications, and recent breakthroughs achieved due to the use of these "omics" techniques.
You have chosen to structure this review according to some topics (taxonomy and classification, food and industry, biomedical sciences), so maybe in the introduction you could for example introduce the overall generalized scheme of "omics" - maybe even slightly modified Figure 1 (and then obviously delete it from the section where it currently appears), then explain what are the "omics", and then only shortly introduce the importance of fungi and the reason for choosing this particular selection of topics? And then within every chapter, try to focus on clearly showing which of the "omics" methods had led to what breakthroughs. As it is now, there is very difficult to see the connection between different information pieces.
Response 6: Thank you for your comment. We have already recorrected as suggestion. In Figure 1, However, the figure solely focuses on the application of omics techniques in fungal omics for ensuring food safety, rather than providing a comprehensive overview of general omic tools. Therefore, to maintain clarity and coherence within this section, it is better to include it as a representation of omics as a tool for ensuring food safety. Furthermore, the figure highlights the significance of utilizing genomics, transcriptomics, proteomics, metabolomics, and ionomics to gain valuable insights into the genetic diversity, gene expression patterns, protein profiles, and metabolite production within food-related fungi. By concentrating on the crucial aspect of food safety, this figure provides a clear and targeted illustration.
Some specific comments:
- a) Lines 80-82 - here you mention the estimates of the fungal diversity, for the first time, already giving some numbers. But then, in lines 192-197 you repeat this information, giving actually additional estimates. I would just leave this information (the more detailed one) in only one place.
Response 7: we have already recorrected as suggestion
- b) While I appreciate some in-depth information provided (e.g. lines 770-816 on antifungal drugs, together with the Figure 2), I feel that this is too much deviating from the fundamental aim of the paper which is the "recent advances of omics". I understand that you have to introduce some of these topics, but I would suggest to try to shorten them, so that they would not dominate the text.
Response 8: we have already shotedted as suggestion
- c) Similar to the previous comment - lines 821-822 "A number of studies on fungal omics data are available, and we recommend that readers consult them for a comprehensive understanding of the utility of fungal omics data [25,70,176,249–251]. In this section, we focus on two types of fungal proteins that have the potential to become important targets for future drug development." - this is extremely confusing - so now you write that this paper is actually not about the "omics", but it will focus on some important proteins from now on... And you even provide the extensive description (lines 825-872) and the figure 3 with the structure of the protein. I do think this is interesting, I do think this is important, but suddenly, instead of clear review showing the application of the "omics", the Reader has a minireview on these proteins! I feel that this has to be contextualized in light of the "omics" methods used to make this discovery. There are several similar issues throughout the manuscript, where you delve too much, in my opinion, into description of the problem, without clearly referring to the role of the "omics".
Response 9: we have already recorrected as suggestion
- d) Please carefully review the information you are giving, and provide citations (including separate citations) for important, information-bearing statements. Some smaller examples:
Response 10: we have already recorrected as suggestion
- Line 60 "Ellis and Sutton [4,5].", should read "...Ellis [4], and Sutton [5]."
Response 11: we have already modified as suggestion
- Line 79 "Mitchell and Zuccaro (2006) and Seifert (2009) [10,11]." - so what's the conception here? Shouldn't it be "Mitchell and Zuccaro [10], and Seifert [11]"?
Response 12: we have already modified as suggestion
- d) You have the paragraph entitled "2.1. Discovering New Taxa in Known Lineages" (line 135), and yet the introductory lines 136-160 are focused on something totally different!
Response 13: we have already modified as suggestion
- e) Please be consistent with omics and "omics" - both versions appear throughout the manuscript.
Response 14: we have already modified as suggestion with all consistency for the whole of ms.
Comments on the Quality of English Language
I really feel that you need some English correction there, and this is the work for English-editing company. This concerns mostly grammar, but also very often the logical meaning of the sentence.
Response 15: we sent the ms for the English Language’s service by English native speaker (AJE; https://www.aje.com) again for make sure all of texts should be corrected.
The revised MS with "Author to respond reviewer" is attached.

Reviewer 4 Report
This manuscript is a work that is focused in to show different applications and research fields of fungal data omics, which can help us to understand the biology, evolution, pathogenicity, and ecology, but also show the potential of these techniques in the exploration of little-studied groups of fungi. I believe that this work can be of interest to researchers in a broad number of disciplines (medical clinicians, taxonomists, biotechnology, and pharmaceutical researchers. The manuscript is clear and focused throughout, however, there are a few suggestions that I believe would enhance this work.
Comments:
There are just a few examples that reflect the application of phylogenomics to improve taxonomic analyzes. I recommend addressing this by adding an example of your choice. Some examples and suggestions are the following:
Li, Y., Steenwyk, J. L., Chang, Y., Wang, Y., James, T. Y., Stajich, J. E., & Rokas, A. (2021). A genome-scale phylogeny of the kingdom Fungi. Current Biology, 31(8), 1653-1665.
Vandepol, N., Liber, J., Desirò, A., Na, H., Kennedy, M., Barry, K., & Bonito, G. (2020). Resolving the Mortierellaceae phylogeny through synthesis of multi-gene phylogenetics and phylogenomics. Fungal diversity, 104(1), 267-289.
Steenwyk, J. L., Balamurugan, C., Raja, H. A., Goncalves, C., Li, N., Martin, F., & Rokas, A. (2022). Phylogenomics reveals extensive misidentification of fungal strains from the genus Aspergillus. bioRxiv, 2022-11.
You can improve the metagenomics subject in metabarcoding and shotgun metagenomics with some examples of applications, ex. Fungal analysis in different environments, dynamics of microbial communities in food, or fungal microbiome response to some diets, or lifestyles.

No comments!
Author Response
Response to Reviewer 4 Comments
This manuscript is a work that is focused in to show different applications and research fields of fungal data omics, which can help us to understand the biology, evolution, pathogenicity, and ecology, but also show the potential of these techniques in the exploration of little-studied groups of fungi. I believe that this work can be of interest to researchers in a broad number of disciplines (medical clinicians, taxonomists, biotechnology, and pharmaceutical researchers. The manuscript is clear and focused throughout, however, there are a few suggestions that I believe would enhance this work.
Response 1: We deeply appreciate your invaluable suggestion.
Comments:
There are just a few examples that reflect the application of phylogenomics to improve taxonomic analyzes. I recommend addressing this by adding an example of your choice. Some examples and suggestions are the following:
Li, Y., Steenwyk, J. L., Chang, Y., Wang, Y., James, T. Y., Stajich, J. E., & Rokas, A. (2021). A genome-scale phylogeny of the kingdom Fungi. Current Biology, 31(8), 1653-1665.
Vandepol, N., Liber, J., Desirò, A., Na, H., Kennedy, M., Barry, K., & Bonito, G. (2020). Resolving the Mortierellaceae phylogeny through synthesis of multi-gene phylogenetics and phylogenomics. Fungal diversity, 104(1), 267-289.
Steenwyk, J. L., Balamurugan, C., Raja, H. A., Goncalves, C., Li, N., Martin, F., & Rokas, A. (2022). Phylogenomics reveals extensive misidentification of fungal strains from the genus Aspergillus. bioRxiv, 2022-11.
Response 2: We sincerely appreciate your invaluable suggestion, and as per your recommendation, we have incorporated the mentioned three citations into our manuscript. Furthermore, we have included on the utilization of phylogenomics to enhance taxonomic analyses in our paper as your suggestion (such as lines 121-142).
You can improve the metagenomics subject in metabarcoding and shotgun metagenomics with some examples of applications, ex. Fungal analysis in different environments, dynamics of microbial communities in food, or fungal microbiome response to some diets, or lifestyles.
Response 3: We greatly appreciate your invaluable suggestion, and as a result, we have enhanced on metagenomics in the context of metabarcoding and shotgun metagenomics in our manuscript. Additionally, we have included some examples of applications to further illustrate our points (such as lines 287-297).
The revised MS with "Response to Reviewer 4 Comments" is attached.

Round 2
Reviewer 1 Report
The manuscript "OMICS and Other Advanced Technologies in Mycological Applications" has been adequately modified.